# A Systematic Review of Intracranial Complications in Adults with Pott Puffy Tumor over Four Decades

**DOI:** 10.3390/brainsci13040587

**Published:** 2023-03-30

**Authors:** Giorgos Sideris, Efstathia Davoutis, Evangelos Panagoulis, Pavlos Maragkoudakis, Thomas Nikolopoulos, Alexander Delides

**Affiliations:** 2^nd^ ENT Department, Attikon University Hospital, 124 62 Athens, Greece

**Keywords:** Pott’s puffy tumor, frontal sinusitis, frontal osteomyelitis, intracranial complication, adults, review

## Abstract

The purpose of this study is to investigate the risk factors of intracranial complications in adult patients with Pott Puffy Tumor (PPT). A systematic review was conducted of clinical studies from January 1983 to December 2022 that reported on PPT adult patients. The full-text articles were reviewed for the patients’ ages, sex, cultured organisms, surgical procedures, clinical sequalae, and underlying diseases that may affect the onset of intracranial complications in PPT adult patients. A total of 106 studies were included. Medical data were reviewed for 125 patients (94 males, 31 females). The median age was 45 years. A total of 52% had comorbidities, mostly head trauma (24.5%), sinus/neurosurgical operations (22.4%), immunosuppression conditions (13.3%), diabetes mellitus (9.1%), cocaine use (7.1%), or dental infections (6.1%). A total of 28 cultures revealed Streptococcus (22.4%), 24 contained staphylococci (19.2%), and 22 cultures contained other pathogens (17.6%). An amount of 30.4% developed intracranial complications, with the most common being epidural abscesses or empyemas (55.3%), as well as subdural (15.7%) and extradural lesions (13.2%). Age, DM, and immunosuppression conditions are significantly associated with intracranial complications (*p* < 0.001, *p* = 0.018 and *p* = 0.022, respectively). Streptococcus infection is associated with intracranial complications (*p* = 0.001), although Staphylococcus and other microorganisms are not. Surgical intervention, mainly ESS, and broad-spectrum antibiotics remain the cornerstones of treatment.

## 1. Introduction

Pott puffy tumor (PPT) is named after Pervical Pott, who initially described this condition in 1768 as a consequence of head trauma and later as a consequence of frontal sinusitis. It refers to the edema and swelling of the forehead and scalp caused by a subperiosteal abscess overlying frontal bone osteomyelitis [1,2].

PPT remains a rare and life-threatening medical condition due to the intracranial extension of the infection, which may spread either directly through the sinus walls (posterior, anterior, or inferior wall) or through the mucosa-draining diploic veins. Intracranial complications occur mainly in adolescents. This is predominantly due to the fact that, during the pneumatization process of the frontal and ethmoid sinuses, the vascularity in the diploic circulation reaches its peak point throughout the adolescent period [3,4].

Several causative factors and comorbidities have been reported as capable of influencing the development of PPT [5]. The most common symptoms include headache, purulent rhinorrhea, fever, frontal sinus tenderness, and periorbital swelling [6,7,8]. Vomiting, photophobia, and other neurological clinical signs of meningitis and encephalitis may represent intracranial complications such as pachymeningitis, frontal lobe abscess, epidural or subdural empyema, and cavernous sinus thrombosis [9,10,11]. While the overall rate of intracranial complications in pediatric and adolescent patients with PPT has been reported to be 72%, several authors indicate that the incidence rate in adults ranges between 29% and 85% [4,12,13].

To date, there has been a lack of studies investigating the prognostic factors for developing intracranial complications in adult PPT patients. This study aims to provide an up-to-date literature review of the epidemiological, clinical, and microbiological findings associated with intracranial complications in PPT adult patients, as well as a discussion of the treatment course, which remains an area of controversy in the literature.

## 2. Materials and Methods

A systematic review of clinical studies reporting PPT in adult patients was conducted following the PRISMA guidelines. The authors independently performed the literature search, study selection, and data extraction. The PUBMED/MEDLINE database was accessed covering the period from 1 January 1983 to 1 December 2022. Our search was limited to articles written in English that used the key terms “Pott’s tumor” and “Pott puffy tumor”. The term “Pott’s tumor” was identified in 308 articles, whereas the term “pott puffy tumor” was identified in 226 articles. The retrieved studies from the initial search were further screened for additional articles (Figure 1).

The main inclusion criterion was that the articles provided the individual characteristics of at least one patient with PPT, aged 17 or older. Cases with orbital or odontogenic involvement were also included. The exclusion criteria involved studies that included children or discussed other causes of frontal bone osteomyelitis. Unrelated articles that were discovered by keyword matching were also excluded. The full-text articles were reviewed for the patients’ ages, genders, cultured organisms, evidence of intracranial complications, surgical procedures, clinical sequalae, and underlying diseases that may affect the onset of PPT and the development of intracranial complications in PPT adult patients.

Qualitative and quantitative analyses of the data (variance of each variable and descriptive statistics measures of central slope and variability) were performed. To determine possible differences between the mean values of the various quantitative variables, the *t* test was used (or its non-parametric equivalent). The x^2^ statistical test was used as a statistical criterion to test the independence between the two studied qualitative variables. Microsoft Excel and IBM SPSS software was used to analyze the data and create the relevant graphs.

## 3. Results

A total of 106 articles regarding PPT in adult patients were included in this study [3,5,6,8,9,11,13,14,15,16,17,18,19,20,21,22,23,24,25,26,27,28,29,30,31,32,33,34,35,36,37,38,39,40,41,42,43,44,45,46,47,48,49,50,51,52,53,54,55,56,57,58,59,60,61,62,63,64,65,66,67,68,69,70,71,72,73,74,75,76,77,78,79,80,81,82,83,84,85,86,87,88,89,90,91,92,93,94,95,96,97,98,99,100,101,102,103,104,105,106,107,108,109,110,111,112]. In total, 125 patients (94 males and 31 females) and their medical records were reviewed. The mean age of all patients was 45.14 ± 18.2 years (95% CI: 49.22–60.84). The minimum age was 17 years and the maximum was 84 years. The mean age of the male and the female patients was 41.88 ± 17.79 and 55.03 ± 15.84, respectively. Age was found to be statistically significant regarding PPT cases between males and females (*p* < 0.001). A total of 65 patients (52%) had comorbidities, the most frequent being head trauma, cocaine use, dental infections, DM, history of sinus/neurosurgical procedures, or causes of immunosuppression. A total of 72 positive cultures (57.6%) were identified out of all the collected swabs and pus cultures. A single pathogen was isolated from 53 patients’ pus or tissue cultures (42.4%); mixed flora were found in 19 patients (15.2%). For the remaining 53 cases, cultures were negative for pathogens, or the authors did not provide data for them (42.4%). *Streptococcus* species were present in 28 cultures (22.4%), Staphylococcus species were found in in 24 cultures (19.2%), while 22 other pathogens were isolated from patients’ cultures (17.6%) (Table 1). 

ENT surgery via an external approach was performed in 26 patients (20.8%); 34 patients (27.2%) underwent endoscopic sinus surgery (ESS) exclusively, while 30 patients (24%) underwent both an external and ESS method. A total of 17 patients (13.6%) underwent neurosurgical procedures, whereas seven patients (5.6%) underwent combination neurosurgical and otolaryngology interventions. Endoscopic sinus surgery (alone or combined) was performed on a total of 68 patients (54.4%). A total of five patients (4%) were treated exclusively with medical antibiotics (Table 2).

Intracranial complications were present in 38 patients (30.4%): 31 males and 7 females. The most common intracranial complication was epidural abscess or empyema (55.3%, 21 patients), followed by subdural empyema (15.7%, 6 patients) and extradural lesions (13.2%, 5 patients). Age was found to be statistically correlated with the development of intracranial complications (*p* < 0.001). Sex, comorbidities, head trauma, dental infection, cocaine use, previous surgical intervention, and the number of pathogens (single pathogen or mixed flora) had no statistical significance between patients with and without intracranial complications. Diabetes mellitus and immunosuppression conditions are significantly associated with intracranial complications (*p* = 0.018 and *p* = 0.022, respectively). There is also a statistical relationship between Streptococcus infection and intracranial complications (*p* = 0.001), while no statistical significance was found between intracranial complications and Staphylococcus or other microorganism infections (Table 3). 

The numbers of patients and intracranial complications are presented for each of the last four decades (Figure 2). 

## 4. Discussion

Our analysis indicates that the number of reported PPT cases in adults and the development of intracranial complications have increased over the past four decades, with almost 50% of them being reported in the last ten years. This may be the result of improved imaging methods and increased awareness of PPT among adults or a possible association with antibiotic resistance in the community [113,114]. In addition, Makery et al. report that adults with recurrent acute rhinosinusitis are more likely to have antibody deficits than children [115]. In spite of the fact that the development of antibiotics contributed to a decline in the frequency of PPT as a result of acute or chronic rhinosinusitis, a number of authors have reported an increase in its prevalence over the past years. However, a broad series of national or population-based studies is lacking.

It is well documented that, concerning adult PPT patients, an extension of the infection in the intracranial cavity may occur through frontal bony erosions, or due to septic emboli via the diploic veins. In the current study, intracranial complications were present in 30% of the patients. Over half of them (55.3%) developed epidural abscess or empyema, while the rest had subdural and extradural lesions. These findings are similar to those that Akiyama et al. report in their review of 32 adult PPT patients, conducted a decade ago [13].

Several researchers suggest a male predominance [13]. We confirm this observation, as the majority of patients are males that are in the third decade of their life, according to our research. In this review, 38 patients developed intracranial complications. Despite the male-to-female ratio of 3:1 and the fact that 80% of the patients with intracranial complications were male, sex was not identified as a contributing factor in the development of intracranial complications. On the other hand, younger patients appear to develop intracranial complications more often. Rohde et al. report that 43.1% of those in their review possessed intracranial pathology, with 69% of the cases being male and 66% of them being under the age of 18. They also report that 78% lacked focal neurologic findings. No direct comparisons can be drawn with the aforementioned study because its authors present data from the literature that also pertain to younger generations [113].

It is not rare for immunocompetent patients with no history of trauma or other predisposing factors and merely common symptoms, such as a swollen forehead and headache, to develop PPT; consequently, one should maintain a high index of suspicion. The establishment of a PPT diagnosis and its possible complications is based on clinical and radiological findings. A CT scan is useful for identifying epidural or subdural abscesses and determining the degree of bone erosion, whereas an MRI scan can detect intracranial involvement [6,97]. It is worth mentioning that 57.1% of the patients included in this study presented with acute or chronic rhinosinusitis, followed in prevalence by head trauma (24.5%), immunosuppression states (13.3%), DM (9.1%), cocaine abuse (7.1%), and dental infections (6.1%). 

As PPT is regarded as a consequence of rhinosinusitis, it is reasonable to discover a significant number of anaerobic isolates in the majority of patients. Anaerobes can exist independently or as a component of diversified flora [22,116]. Although it is reported that patients with PPT typically have polymicrobial involvement in their cultures, mixed flora were isolated in only 26% of our sample [6]. Our data suggest that Streptococcus appears to be the predominant causal microorganism (present in 39% positive cultures), followed by the Staphylococcus species (33%). Other microorganisms (bacteroeides, Hib, micromonas, fungi, pseudomonas aeruginosa, etc.), either alone or as part of a mixed flora, were identified from 44% of our sample cultures. Thus, our findings confirm the previous literature recommendations for intravenous antibiotics, i.e., that they should (a) penetrate the blood–brain barrier; (b) cover both aerobic and anaerobic bacteria; (c) be based on culture results; and (d) be administered over the course of six weeks [6,116,117]. After all, in most PPT cases, streptococci, staphylococci, and anaerobic bacteria are the most commonly isolated species due to the decreased oxygen content in the frontal sinus [6,9].

Streptococcus was the only microorganism that was related to intracranial complications in adult patients with PPT, as it was isolated in 42% of these patients (16 out of 38). Moreover, the odds of the appearance of intracranial complications are 2.3 times greater in patients with streptococcal infections. This is not unexpected, given that previous researchers have observed an increase in the prevalence of Streptococcus species in individuals with acute complex sinusitis [118].

Comorbidities were present in 66.3% of the patients. None of the comorbidities, except DM and immunosuppression states, were statistically correlated with the development of intracranial complications. This observation leads to the consideration that adult PPT patients with a medical history of head trauma, cocaine use, previous dental infections, or other immunosuppressions may not have a higher risk of intracranial complications, contrary to previous reports in the literature. On the other hand, patients with DM or other immunosuppression diseases seem to have 2.3 times the risk of developing intracranial complications compared to patients with no such medical history. 

There are a range of treatment options described in the literature. Guidelines for treatment plans and the proper timing of action have not been sufficiently examined. Several authors claim that a supraorbital rim or glabellar incision is sufficient for abscess drainage, and that it should be followed by endoscopic ethmoidectomy and frontal sinusotomy [6]. Others report that the need for surgical treatment depends on the severity of the infection, with intracranial extension signaling the need for emergency craniotomies and frontal sinus cranialization after total sequestrum removal [89]. On the other hand, it has been suggested that endoscopic frontal sinusotomy (Draf IIa) is a viable and safe approach for the surgical management of the vast majority of PPT patients and that a burr hole is appropriate in the event of intracranial complications [77]. In the review by Akiyama et al., external surgical procedures were chosen in 58.1% of cases, while endoscopic sinus surgery was chosen in 32.9% of cases, despite the need for external subperiosteal abscess drainage in all of these cases [13]. The fact that intracranial extension does not always require neurosurgical intervention has also been reported by other authors, as some complications are often asymptomatic when the abscess is found in silent areas of the central nervous system [6,96].

In recent years, ESS has been performed in combination with simple percutaneous drainage with positive results. Our review shows that external drainage was performed in 44.8% of the cases; 53.3% of them were performed in conjunction with an ESS, while ESS alone was performed in 27.2% of the total cases presented. Moreover, ESS has been the cornerstone of surgical treatment in the last several years, being utilized alone or in combination for 54% of patients. 

The optimal time to perform radical drainage surgery on adult PPT patients has not been determined, and further research is required to identify whether immediate radical surgery reduces the risk of intracranial complications. Nevertheless, our data lead us to the opinion that the presence of intracranial complications, neurological clinical signs, and worsening of common symptoms 48 h after intravenous antibiotic therapy indicates the need for immediate surgical drainage (with an external or endoscopic approach). In the majority of PPT cases (those with gradual resolution of symptoms and no intracranial complications), treatment with iv broad-spectrum antibiotics and, if needed, simple drainage should be a reasonable choice, and surgical intervention should be delayed and performed in a surgical field free of inflammation. Nevertheless, surgical intervention, broad-spectrum antibiotics, and treatment of the underlying disease, especially DM and immunosuppression conditions, represent the cornerstones of treatment in adults with PPT, while multidisciplinary management with neurosurgical and ophthalmology assessment is always recommended. Cases of recurrence have not been reported in the literature.

Despite the fact that this is the first study to provide the risk factors of intracranial problems in adult PPT patients, the majority of the included articles are case reports. Thus, the analysis presented has limitations, since a meta-analysis is not appropriate. Furthermore, our claim that the prevalence of PPT in adults has increased over the studied period is not sufficiently supported, as the increase may simply reflect an increase in the likelihood that these publications will occur. It is also possible that cases with intracranial complications were more likely to be published than those without, given the severity of the disease.

## 5. Conclusions

The reported cases of adult patients with PPT have increased during the past decade. Over 30% of PPT cases may develop intracranial complications, the most common of which are epidural abscesses or empyemas. Intracranial complications are more common in adult males in the fourth decade of their lives. Diabetes mellitus and immunosuppressive conditions are independent prognostic factors for the development of intracranial complications. Streptococcus is identified as the most prevalent pathogen and the only one linked to an increased risk of intracranial complications. Surgical intervention and broad-spectrum antibiotics remain the cornerstones of treatment. A combination of ESS and antibiotic treatment can be effective in the majority of adult PPT cases.

## Figures and Tables

**Figure 1 brainsci-13-00587-f001:**
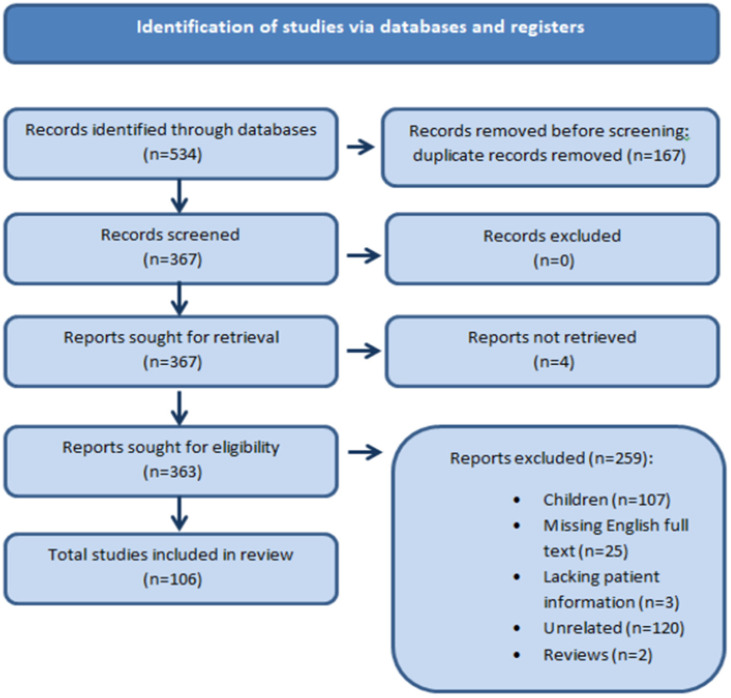
PRISMA flow diagram of the search for PPT.

**Figure 2 brainsci-13-00587-f002:**
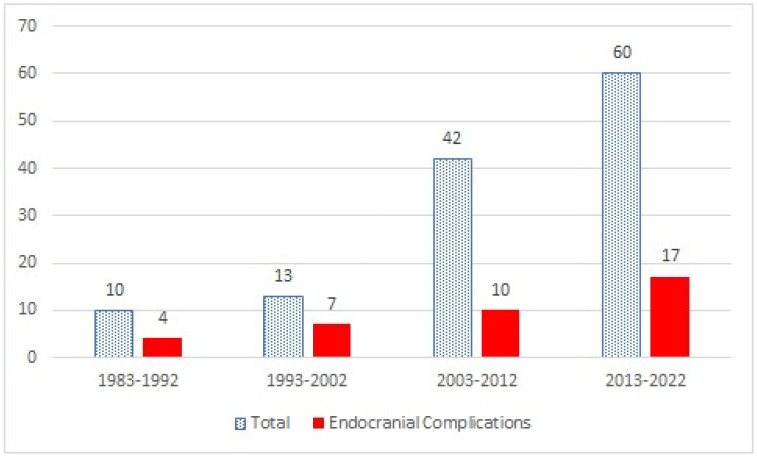
Bar diagram showing reported patients and intracranial complications by decades (1983–2022).

**Table 1 brainsci-13-00587-t001:** Clinical characteristics of adult PPT patients from 1983.

			Total (%) n = 125 Patients
Sex	Male		94 (75.2)
	Female		31 (24.8)
Age (years)	Mean ± SD		45.14 ± 18.2
	95% CI		49.22–60.84
Microbiology (microorganisms identified) ^1^	Yes		72
		Single	53 (73.6)
		Mixed	19 (26.4)
	None		12
	N/A		41
Intracranial complications	Yes		38
	No		87
Comorbidities (in total) ^2^	Yes		65
		Head Trauma	24 (24.5)
		Dental Infection	6 (6.1)
		Cocaine	7 (7.1)
		Diabetes Mellitus	9 (9.1)
		Immunosuppression	13 (13.3)
		Previous ESS, N/S intervention	22
	No		33
	N/A		27

^1^ Pathogens: *Streptococcus* spp., *Staphylococcus* spp., *Bacteroides*, *Proteus* spp., *Hemophilus influenzae*, *Mycobacterium tuberculosis*, *Eubacterium lentum*, *Fusobacterium*, *Propionibacterium*, *Pasteurella multocida*, *Pseudomonas aeruginosa*, *Escherichia coli*, *Aspergillus flavus*, *Mucor*, *Candida Albicans*, *Prevotella* spp., *Peptostreptococcus* spp., *Micromonas micros*, *Prevotella orallis*, *Peptostreptococcus*, *Mycoplasma*, *Enterococcus faecium*, *Collinsella aerofaciens*, *Actinomyces species*, *Bacteroides*, *Stenotrophomonas maltophilia*, *Actinomyces naeslundi*, *Achromobacter xylosoxidans*, *Corynebacterium* spp. N/A: Not applicable. ^2^ Comorbidities: head trauma; dental infection; insect bite; intranasal and inhaled methamphetamine or cocaine; previous endoscopic sinus surgery (ESS) or neurosurgical (N/S) intervention; diabetes mellitus (DM); chronic renal failure (CRF); chronic rheumatoid arthritis (CRA); psoriatic arthritis (PA); human immunodeficiency virus (HIV); chronic alcoholism with related cirrhosis and hepatitis; breast cancer with metastasis; aplastic Anaimia (AA); pregnancy, psychiatric disorders; mitral valve replacement; COVID-19-related mucormycosis; asthma. N/A: not applicable.

**Table 2 brainsci-13-00587-t002:** Treatment course and surgical management.

	Otolaryngology Procedures		Neurosurgical Procedures ^3^	Antibiotics
	External approach ^1^	Endoscopic sinus surgery ^2^		
Only	26	34	17	5
Combined	30		7	114 (plus surgery) ^4^

^1^ External approach: external frontoethmoeidectomy, trephination, Caldwell-Luc, external drainage (unspecified approach), external incision, needle aspiration, obliteration of sinus, osteoplastic flap. ^2^ Endoscopic sinus surgery: endoscopic frontoethmoeidectomy, Draf IIa, IIb, III frontal sinus surgery, balloon sinuplasty. ^3^ Craniectomy, craniotomy. ^4^ Six patients had unspecified treatment courses.

**Table 3 brainsci-13-00587-t003:** Correlation of demographics, medical history, and microbiology findings between adult PPT patients with and without intracranial complications.

		Intracranial Complications n = 38	No Intracranial Complications n = 87	*p* Value
Age (years)	Mean ± SD	35.39 ± 17.70	49.40 ± 16.77	<0.001
Sex	Male	31	63	0.28
	Female	7	24	
Comorbidities (in total) ^1^		21	44	0.84
Head Trauma ^1^		5	19	0.19
Dental infection ^1^		2	4	0.93
Cocaine use^1^		3	4	0.67
Diabetes Mellitus ^1^		6	3	0.02
Immunosuppression ^1^		8	5	0.02
Previous surgery	ESS	5	0	0.18
	Neurosurgery	5	1	0.61
	ESS + Neurosurgery	10	1	0.17
Pathogens ^2^	Single	18	35	0.3
	Mixed flora	9	10	
	*Streptococcus* spp.	16	12	<0.006
	*Staphylococcus* spp.	7	17	0.3
	Other	5	17	0.08

^1^ Only those studies that clearly reported whether or not comorbidities, head trauma, dental infection, cocaine use, diabetes mellitus, or other immunosuppression causes were present are included, n = 98 patients. ^2^ Only those studies that clearly reported which and how many pathogens were isolated or not were included, n = 72 patients.

## Data Availability

Not applicable.

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
