# Peer review of "A Systematic Review of Intracranial Complications in Adults with Pott Puffy Tumor over Four Decades"

_brainsci, 2023, doi:10.3390/brainsci13040587_

Round 1

Reviewer 1 Report

Authors present a systematic review on Pott´s puffy tumor intracranial complications in adult patients. Out of 125 patients from the literature, 30% developed intracranial complications.   Age, DM and immunosuppression conditions as well as infection with Streptococcus were associated with intracranial complications. For a systematic review, we would expect a figure, photo or a radiological imaging of this condition. A table containing 125 patients is too much and I suggest to shorten it. Are these complications complications of PPT OR complications of its intracranial treatment? Instead of providing a table over 4 pages, I suggest to include in the results the treatment modality. Some important surgical aspects are missing and not discussed at all - In a case of cerebral empyema, usually one would remove the bone flap. How was this condition actually treated? Removal of the bone flap, craniotomy? What are the surgical principles and thoughts? Is there any prevention or preventive measures of intracranial spread? For how long was the antibiotic therapy administered, were there any recurrences? 

Author Response

Response to Reviewer 1

For the manuscript entitled “A Systematic Review of Intracranial Complications in Adults with Pott Puffy Tumor Over Four Decades

On behalf of the authors, we would like to thank you for your useful comments.

  • For a systematic review, we would expect a figure, photo or a radiological imaging of this condition”: DONE (figures and images are included in the Graphical Abstract). If it is considered necessary we can include them in the body of Manuscript
  • “A table containing 125 patients is too much and I suggest to shorten it”: DONE
  • “Are these complications complications of PPT OR complications of its intracranial treatment?” Our response: These are reported intracranial complications of PPT”
  • “Instead of providing a table over 4 pages, I suggest to include in the results the treatment modality; What are the surgical principles and thoughts? Is there any prevention or preventive measures of intracranial spread? For how long was the antibiotic therapy administered, were there any recurrences?” We addressed these comments by adding: “There are a range of treatment options described in the literature. Guidelines for treatment plans and the proper timing of action have not been sufficiently examined. Several authors claim that a supraorbital rim or glabellar incision is sufficient for abscess drainage, followed by endoscopic ethmoidectomy and frontal sinusotomy [6]. Others report that surgical treatment depends on the severity of the infection, with intracranial extension signaling the need for emergency craniotomies and frontal sinus cranialization after total sequestrum removal [89]. On the other hand, it is suggested that endoscopic frontal sinusotomy (Draf IIa) is a viable and safe approach for the surgical management of the vast majority of PPT patients and that a burr hole is appropriate in the event of intracranial complications [77]. In the review by Akiyama et al., external surgical procedures were chosen in 58.1% of cases, while endoscopic sinus surgery was chosen in 32.9% of cases, despite the need for external subperiosteal abscess drainage in all of these cases [13]. The fact that intracranial extension does not always require neurosurgical intervention has also been reported by other authors as some complications are often asymptomatic when the abscess is found in silent areas of the central nervous system [6,96]. In recent years, ESS has been performed in combination with simple percutaneous drainage, with positive results. Our review shows that external drainage was performed in 44.8% of the cases, 53.3% of them in conjunction with an ESS, while ESS alone was performed in 27.2% of the total cases presented. Moreover, ESS has been the cornerstone of surgical treatment in the last several years, being utilized alone or in combination in 54% of patients. The optimal time to perform radical drainage surgery on adult PPT patients has not been determined, and further research is required to identify whether immediate radical surgery reduces the risk of intracranial complications. Nevertheless, our data leads us to the opinion that the presence of intracranial complications, neurological clinical signs, and worsening of common symptoms 48 hours after intravenous antibiotic therapy indicates the need for immediate surgical drainage (with an external or endoscopic approach). In the majority of PPT cases (those with gradual resolution of symptoms and no intracranial complications), treatment with iv broad-spectrum antibiotics and, if needed, simple drainage should be a reasonable choice, and the surgical intervention should be delayed and performed in a surgical field free of inflammation. Nevertheless, surgical intervention, broad-spectrum antibiotics, and treatment of the underlying disease, especially DM and immunosuppression conditions, represent the cornerstones of treatment in adults with PPT, while multidisciplinary management with neurosurgical and ophthalmology assessment is always recommended. Cases of recurrence have not been reported in the literature. Despite the fact that this is the first study to provide the risk factors for intracranial problems in adult PPT patients, the majority of the included articles are case reports. Thus, the analysis presented has limitations, since a meta-analysis is not appropriate. Furthermore, our claim that the prevalence of PPT in adults has increased over the studied period is not sufficiently supported, as the increase may simply reflect an increase in the likelihood that these publications will occur. It's also possible that cases with intracranial complications were more likely to be published than those without, given the severity of the disease”.

Reviewer 2 Report

The authors report a systematic review of Pott's Puffy Tumor (PPT). They have searched the literature and have identified 125 cases, from 106 case reports or case series, which constitutes the population studied in their review. Based on these cases, they conclude that the prevalence of PPT has increased over the last decade (although their aim stated that they wished to determine the incidence of PPT - not the same thing). They also identify risk factors for complications of the disease. 

The methodology used cannot identify either the incidence or prevalence of the disease. While the number of cases reported has increased over time, numerous reasons for that increase unrelated to incidence or prevalence are possible, including more cases because of a larger population, increased pressure for publication so more cases are reported, and changes in diagnostic or surgical techniques warranting publication of new knowledge. 

Additionally, the INTRODUCTION summarizes known information about PPT, rather than laying a foundation for why the study is warranted. Does collating the collection of cases in this review address some controversy or potential gap in understanding that makes the review a worthwhile contribution?

A few additional points for the authors to address follow:

1.  In the ABSTRACT on line 15, what is meant by the statement "106 studies participated?"

2.  In METHODS, why were children under 17 years old excluded?

3.  In RESULTS, line 93, regard the relationship between age and sex is worded poorly. "Incidence" has not been established. Perhaps the authors meant that female patients are older than male patients.

4.  In RESULTS, line 153, incidence is not increased. Rather, the number of reports of PPT has increased.

Author Response

Response to Reviewer 2

For the manuscript entitled “A Systematic Review of Intracranial Complications in Adults with Pott Puffy Tumor Over Four Decades

On behalf of the authors, we would like to thank you for your useful comments.

  • “The methodology used cannot identify either the incidence or prevalence of the disease. While the number of cases reported has increased over time, numerous reasons for that increase unrelated to incidence or prevalence are possible, including more cases because of a larger population, increased pressure for publication so more cases are reported, and changes in diagnostic or surgical techniques warranting publication of new knowledge”. We address this comment by adding a paragraph at the end of the discussion section: Despite the fact that this is the first study to provide the risk factors for intracranial problems in adult PPT patients, the majority of the included articles are case reports. Thus, the analysis presented has limitations, since a meta-analysis is not appropriate. Furthermore, our claim that the prevalence of PPT in adults has increased over the studied period is not sufficiently supported, as the increase may simply reflect an increase in the likelihood that these publications will occur. It's also possible that cases with intracranial complications were more likely to be published than those without, given the severity of the disease”. Moreover we replaced the word “incidence” with the word “cases” in our manuscript.
  • “The INTRODUCTION summarizes known information about PPT, rather than laying a foundation for why the study is warranted. Does collating the collection of cases in this review address some controversy or potential gap in understanding that makes the review a worthwhile contribution?” We changed the Introduction section according to your comments trying to make clear the main research topic. Thus we reduced original text and added two new paragraphs and one new reference: FIRST PARAGRAPH ADDED: PPT remains a rare and life-threatening medical condition due to the intracranial extension of the infection, which may spread either directly through the sinus walls (posterior, anterior or inferior wall) or through the mucosa-draining diploic veins. In-tracranial complications occur mainly in adolescents and predominantly due to the fact that during the pneumatization process of the frontal and ethmoid sinuses, the vascularity in the diploic circulation reaches its peak point throughout the adolescent period [3-4] ....... SECOND PARAGRAPH ADDED:While the overall rate of intracranial complications in pediatric and adolescent patients with PPT has been reported to be 72%, several authors indicate that the incidence rate in adults ranges between 29% and 85% [4,12-13]. To date, there are a lack of studies investigating the prognostic factors for developing intracranial complications in adult PPT patients. This study aims to provide an up-to-date literature review of the epidemiological, clinical, and microbiological findings associated with intracranial complications in PPT adult patients as well as a discussion of the treatment course, which remains an area of controversy in the literature”.
  • “In the ABSTRACT on line 15, what is meant by the statement "106 studies participated?" Corrected to “106 studies were included”
  • “In METHODS, why were children under 17 years old excluded?” Our response:PPT occurs mainly in adolescents and predominantly due to the fact that during the pneumatization process of the frontal and ethmoid sinuses, the vascularity in the diploic circulation reaches its peak point throughout the adolescent period. In the aforementioned age range, there is also a higher incidence of acute bacterial sinusitis and upper respiratory tract infections. Moreover, most countries, including the authors', have a 16-year age limit between adults and pediatric patients”
  • “In RESULTS, line 93, regard the relationship between age and sex is worded poorly. "Incidence" has not been established. Perhaps the authors meant that female patients are older than male patients.” The word “incidence” was replaced by the word “cases”
  • “In RESULTS, line 153, incidence is not increased. Rather, the number of reports of PPT has increased”. DONE

Round 2

Reviewer 1 Report

Authors have sufficiently responded to reviewer remarks. 

Reviewer 2 Report

The authors have satisfactorily addressed the reviewer's comments.